# The Value of Sport: Wellbeing Benefits of Sport Participation during Adolescence

**DOI:** 10.3390/ijerph19148579

**Published:** 2022-07-14

**Authors:** Oliver W. A. Wilson, Chris Whatman, Simon Walters, Sierra Keung, Dion Enari, Andy Rogers, Sarah-Kate Millar, Lesley Ferkins, Erica Hinckson, Jeremy Hapeta, Michael Sam, Justin Richards

**Affiliations:** 1Te Hau Kori, Faculty of Health, Victoria University of Wellington, Wellington 6140, New Zealand; oliver.wilson@vuw.ac.nz; 2School of Sport & Recreation, Auckland University of Technology, Auckland 1011, New Zealand; chris.whatman@aut.ac.nz (C.W.); simon.walters@aut.ac.nz (S.W.); sierra.keung@aut.ac.nz (S.K.); dion.enari@aut.ac.nz (D.E.); lesley.ferkins@aut.ac.nz (L.F.); erica.hinckson@aut.ac.nz (E.H.); 3Sport New Zealand, Wellington 6140, New Zealand; andy.rogers@sportnz.org.nz; 4Faculty of Health, University of Canterbury, Christchurch 8140, New Zealand; sarah-kate.millar@canterbury.ac.nz; 5School of Physical Education, Sport and Exercise Sciences, University of Otago, Dunedin 9016, New Zealand; jeremy.hapeta@otago.ac.nz (J.H.); mike.sam@otago.ac.nz (M.S.)

**Keywords:** physical activity, sport, exercise, recreation, leisure, wellbeing, happiness, youth, young people

## Abstract

Insight into the unique benefits of sport participation above and beyond those associated with participation in other physical activities among adolescents is limited in Aotearoa New Zealand (NZ). The purpose of this study was to examine the association between wellbeing and organised sport participation among adolescents whilst accounting for demographic characteristics and other recreational physical activity. Demographic characteristics (age, gender, ethnicity, deprivation, (dis)ability status), organized sport, recreational physical activity, and wellbeing were assessed in cohorts of NZ adolescents (11–17 years) between 2017 and 2019. After adjusting for demographics, better wellbeing was associated with participation in any recreational physical activity (OR = 2.49, 95%CI = 1.97–3.13), meeting physical activity recommendations (OR = 1.63, 95%CI = 1.47–1.81), and each additional hour of recreational physical activity (OR = 1.03, 95%CI = 1.02–1.04). After adjusting for demographics and overall recreational physical activity participation, better wellbeing was also associated with participation in any organized sport (OR = 1.66, 95%CI = 1.49–1.86), and each additional hour of organized sport (OR = 1.09, 95%CI = 1.07–1.11). Although participation in recreational physical activity appears to be beneficial for wellbeing, organized sport appears to offer unique additional wellbeing benefits. Positive experiences of organized sport participation may offer additional wellbeing value above and beyond other recreational physical activity types in young people who are active.

## 1. Introduction

The promotion of national wellbeing is acknowledged as a priority internationally [1,2,3], including in Aotearoa New Zealand (NZ) where child and youth wellbeing in particular is increasingly being prioritized [4,5]. While there is ongoing conjecture concerning exactly what defines wellbeing internationally, in the NZ context, NZ’s Living Standards Framework (LSF) clearly sets forth 12 domains that comprise individual and collective wellbeing and the roles of institutions and organisations in facilitating wellbeing [6]. The development of the LSF was informed by decades of international wellbeing research [7].

There is an emerging evidence base supporting the wellbeing value of quality physical activity experiences. Numerous studies have indicated that there is a positive association between subjective wellbeing and physical activity [8,9]. Evidence among adolescents also suggests that there is a reciprocal relationship between physical activity and subjective wellbeing [10]. Beyond subjective wellbeing, there is a wealth of evidence demonstrating an association between physical activity and constructs related to wellbeing among children and adolescents, including: mental [11,12,13,14,15] and physical health [16]; social connections and support [17,18,19,20,21]; cognition, academic achievement, and physical literacy [12,22,23,24,25]; socio-economic status [26]; and, housing, environmental amenity, and safety [27,28,29,30,31].

Continuing to develop an understanding of the potential and nuanced contribution of physical activity and sport to population wellbeing is indicated. Evidence indicates that recreational physical activity may have an additional benefit to wellbeing beyond other domains of physical activity [32]. Similarly, sport participation may have benefits that are unique when compared to other forms of recreational physical activity [33]. However, limited conclusive evidence examining the contributions of different types of physical activity to wellbeing among adolescents is available [34], particularly in the context of NZ. Mixed findings from recent research concerning sport and wellbeing in NZ indicate that further investigation to better understand this relationship is warranted [35]. 

Thus, the aim of this study was to examine how wellbeing is associated with overall recreational physical activity and organised sport participation in young people, whilst accounting for a broad range of relevant socio-demographic characteristics [36]. This will provide an insight into the potential unique contribution to the wellbeing of young people of participation in organised sport vs. recreational physical activity more generally. In doing so, findings will offer an insight into whether quality sport experiences have an additional wellbeing benefit in a NZ context, as well as offer guidance as to what types of physical activity should be prioritised to optimise its contribution to the wellbeing of young people in NZ.

## 2. Materials and Methods

### 2.1. Participants and Procedures

Data were collected as a part of the Active NZ Young Peoples survey [37]. Data included in the current study were collected continuously from the beginning of 2017 to the end of 2019. Young people, children and adolescents aged 5–17 years at baseline, were recruited via adults residing in their household who were identified to participate in the Active NZ adults survey using the NZ electoral roll as a sampling frame. Full survey methods are detailed in the annual Active NZ Technical reports [38,39,40]. Participants who did not have complete socio-demographic characteristics, physical activity, and wellbeing data were excluded. Those younger than 11 years and those no longer at school were also excluded. The final analyses included data for 6771 young people.

### 2.2. Measures

#### 2.2.1. Demographic Characteristics

Age: Participants identified their age in years.

Gender: Participants identified their gender (male, female, or gender diverse). Due to limited sample size for gender diverse our inferential analyses focused on cis-gender individuals.

Ethnicity: Participants identified their ethnic group(s), and there was no limit on the number of ethnicities they could choose. For the purposes of these analyses, participants who identified multiple ethnicities were categorised to only one ethnic group using the following prioritisation: Māori, Pasifika, Asian, Middle Eastern/Latin American/African (MELAA), European, other. These ethnic groups were selected based on those specified by Statistics NZ. Due to limited sample size for other ethnicities our inferential analyses did not include this group.

Disability status: Participants who did not report using a wheelchair, using a walking aid, using prosthetics, or dealing with an ongoing physical illness were classified as someone without a disability.

Deprivation status: Deprivation was determined using the 2018 NZ Index of Deprivation, which combines census data relating to income, home ownership, employment, qualifications, family structure, housing, access to transport and communications to designate small geographic areas (60–110 people) with a decile number ranging from 1 (least deprived) to 10 (most deprived) [41]. Participants were classified as residing in low (deciles 1–3), medium (deciles 4–7), and high (deciles 8–10) deprivation areas.

#### 2.2.2. Physical Activity and Sport Participation

Participation: Participants were asked whether they had performed any physical activity that was specifically for the purpose of sport, exercise, or recreation in the past seven days (yes/no). 

Those who answered yes were classified as participants in “recreational physical activity” and were then asked to identify from a list of 77 options which activities they participated in during the past seven days. There was also an “other” option provided with free text for participants to describe any activity they had performed that was not listed.

Setting: For activities that they had participated in, participants were asked in what settings they had participated in (“in PE or class at school”, “in a competition or tournament”, “training or practicing with a coach/instructor”, “playing or hanging out with family or friends”, “playing on my own”, or “for extra exercise, training, or practice without a coach or instructor”).

Duration: If participants indicated that they had participated in a given activity in a given setting they were asked how long they participated in the activity/setting in a given week (15 min, 30 min, 45 min, 1 h, 1.5 h, 2 h, 3 h, 4 h, or 5 h of more).

Physical activity and sport classification: The list of recreational physical activities included non-sport recreational activities (e.g., tramping or bush walks) and exercise (e.g., gym), as well as a range of sports. For the purpose of this study, the following activities were considered “sport physical activity”: Adventure racing, athletics, badminton, basketball, body boarding, boxing, canoeing or kayaking, cheerleading, cricket, croquet, cross country, cycling of biking, dance/dancing, football/soccer, futsal, golf, gymnastics, handball, hockey or floorball, indoor climbing, jiu jitsu, ki-o-rahi, kapa haka, karate, mountain biking, motorbiking, motocross, netball, orienteering, paddle boarding, parkour, rock climbing, rollerblading, roller skating, rowing, rugby or rippa rugby, rugby league, running/jogging, sailing or yachting, scuba diving, scootering, skateboarding, skiing, snowboarding, softball, squash, surf lifesaving, surfing, swimming, table tennis, taekwondo, tennis, touch, trampoline, triathlon or duathlon, ultimate frisbee, volleyball, waka ama, wake boarding, water polo or flippa ball, water skiing. 

Recreational physical activity and organized sport definitions: Several recreational physical activity and organized sport variables were included in analyses in the current paper. These variables were defined as follows:Physically active—participation in any recreational physical activity (active vs. inactive)Recreational physical activity duration—sum of durations (hours/week) across all listed activities and settingsMeeting physical activity recommendations—≥420 min/week of recreational physical activity (meeting recommendations vs. not meeting recommendations) [42].Organized sport participant—participation in any sport physical activity “in a competition or tournament” and/or “training or practicing with a coach/instructor” (participant vs. non-participant)Organized sport activity duration—sum of durations (hours/week) for sport physical activity “in a competition or tournament” and/or “training or practicing with a coach/instructor”.

#### 2.2.3. Wellbeing

Participants were asked to respond to a question rating their wellbeing on a 10-point scale ranging from 1 (very unhappy) to 10 (very happy). Whilst it is recognized that wellbeing is a multi-dimensional construct, the single item measure used in this study has been shown to be a valid overall wellbeing indicator and aligns with the OECD Guidelines on Measuring Subjective Wellbeing [43]. Based on the distribution of the data, participants whose response was ≥8 were categorized as having “better wellbeing”. 

### 2.3. Statistical Analyses 

Analyses were conducted using SPSS (Version 28.0, IBM, Armonk, NY, USA). Descriptive statistics were computed to describe the sample. Binary logistic regression analyses were conducted to examine the association between wellbeing and the various recreational physical activity and organized sport variables. Two different analyses were conducted for the association between wellbeing and the physical activity and organized sport variables: Model 1 was a crude unadjusted model; Model 2 was adjusted for socio-demographic characteristics. A third model was completed for the organized sport variables, which adjusted for socio-demographic characteristics and total recreational physical activity duration. We calculated 95% confidence intervals (CIs) for all of the odds ratios (ORs) reported and used these to assess statistical significance (i.e., 95% CIs not crossing 1.0 equivalent to *p* < 0.05).

## 3. Results

### 3.1. Participant Characteristics

Participant characteristics are reported in Table 1. The sample was relatively evenly split between males and females, and the majority were European (58.5%), were without a physical disability (94.7%), and resided in low–mid-deprivation areas (77.7%). Nearly all of the sample were active, i.e., reported participating in some physical activity (94.7%). The average duration of recreational physical activity participation was 10.9 ± 10.1 h/week and most of the sample reported participating in sufficient physical activity to meet physical activity recommendations (58.4%). The average duration of organized sport participation was 2.8 ± 3.6 h/week, with most of the sample reportedly participating in organized sport (63.5%). Most participants were categorized as having good wellbeing (63.0%), with an average response to the wellbeing item of 7.7 ± 1.7.

### 3.2. Association between Physical Activity/Organized Sport Participating and Wellbeing

All the physical activity and organized sport variables have a significant positive association with wellbeing in the crude model (Model 1). The results from Model 2 indicate that adolescents that do any recreational physical activity have 2.49 higher odds of having better wellbeing than those who do no recreational physical activity. Those who met physical activity recommendations had 63% higher odds of having better wellbeing than those below this threshold. The odds of having better wellbeing were also 3% higher for every additional hour of participation in any recreational physical activity. The results from Model 3 indicate that participation in organized sport was associated with 66% higher odds of having better wellbeing, independent of total recreational physical activity participation. Every additional hour of organized sport participation was associated with 9% higher odds of having better wellbeing, independent of total recreational physical activity participation (Table 2).

## 4. Discussion

Our results indicate that participating in recreational physical activity is positively associated with wellbeing during adolescence in NZ. Young people who do any recreational physical activity are more likely to have better wellbeing and there appeared to be additional benefit for each additional hour of participation. However, our findings also suggest that participation in organised sport was even more strongly associated with wellbeing outcomes for young people in NZ, even after taking into account total duration of recreational physical activity participation.

The positive association between physical activity participation and wellbeing among adolescents is consistent with previous research, which has also identified several potential neurobiological, psychosocial, and behavioural pathways for this relationship [10,12,44]. Our findings suggest that any recreational physical activity participation is better than none, and that there is a positive dose–response relationship. This also aligns with the dose–response curve observed in previous research examining the association between wellbeing and physical activity among adults [9]. The cross-sectional nature of our study prevents determination of the direction of causation for the associations between physical activity participation and wellbeing. Although there is strong evidence regarding the impact of physical activity participation on wellbeing [11,15,33,45,46,47,48], a reciprocal relationship is probable [10]. This means that while physical activity participation improves wellbeing it is also likely that better wellbeing facilitates greater physical activity participation (i.e., a virtuous cycle). Thus, beyond advocating for physical activity and sport to promote youth wellbeing, fostering youth wellbeing using other means could also directly contribute to enhancing participation in physical activity.

Our findings also indicate that participation in organized sport offers a unique benefit to wellbeing above and beyond participation in other recreational physical activities. This is consistent with the conclusions of a prior systematic review concerning the benefits of participation in sport for children and adolescents [33]. It is worth noting that the magnitude of the apparent benefit from additional participation in organized sport is considerably larger than that of additional participation in overall recreational physical activity in our study. This was the case for participating in any organized sport (i.e., vs. none) and for each additional hour of participation. That being said, it is well established that there is a limit beyond which the impact on wellbeing of additional participation in organized sport plateaus and may actually start to decrease. This is particularly pertinent when participation is driven by early specialization, which can contribute to burnout and musculoskeletal injuries stemming from overuse [49,50,51]. We were not able to examine this in our analyses due to limitations in the physical activity duration data available.

Although examining the mechanisms that explain why sport may offer benefits to wellbeing above and beyond participation in other recreational physical activities is beyond the scope of our study, we can surmise several hypotheses from the existing literature. Positive sporting experiences may provide young people with a better opportunity to realize benefits stemming from social connections and a sense of relatedness, competence, and achievement. The organized sport context in NZ is widely recognized as a space that aims to facilitate both bonding and bridging social capital in local communities [52]. There is also evidence from studies of young adults suggesting that more intrinsic motives (enjoyment and challenge) are associated with sport, whereas more extrinsic motives (appearance, weight, and stress management) are associated with exercise [53]. Indeed, evidence suggests that intrinsic motivation, perceived competence, and relatedness tend to be higher among adolescents who participate in sporting activities compared to those who participate in non-sporting physical activities or are inactive [54]. Positively influencing these interpersonal and intrapersonal characteristics are explicitly recognized as key objectives in the coach development pathways for numerous sports in NZ [55]. Given the prominence of “coaches” in how we have defined organized sport in this study, it is likely that experiences with sport coaches have directly contributed to our wellbeing findings.

The current study is not without limitations beyond its cross-sectional design. Self-report measures of physical activity tend to overestimate activity levels [56]. However, given our focus on physical activity behaviour (i.e., type of activity), rather than on duration of movement (i.e., device-based measures), self-report methods are the most pragmatic and valid way to collect data from an adequate sample as in our study. There are also limitations in the way we have measured wellbeing. Although the single item we used does not encompass all of the domains of wellbeing outlined in the LSF, such single items have been shown to be valid and robust measures of overall wellbeing internationally [57]. However, it is unknown how well the wellbeing single item we used captures the wellbeing of Māori and/or Pacific people in NZ. Wellbeing described by these population groups emphasizes interpersonal relationships (particularly whānau and family), culture, religion, connectedness, belonging, and geographical dimensions [58,59,60,61]. Consequently, further research is warranted to understand the relevance of our findings in these population groups and more broadly across all of the wellbeing domains outlined in the LSF and other constructs of wellbeing for different population groups.

## 5. Conclusions

In summary, participation in organized sport appears to offer a unique benefit to wellbeing above and beyond participation in other recreational physical activities. Thus, while quality experiences of recreational physical activity are evidently beneficial for wellbeing, promoting participation in organized sport may offer greater value for those who are already active. Further investigation into whether the wellbeing benefits of sport vary based on setting and/or type of sporting activity is warranted, as is further research on understanding the mechanisms that underpin why sport may offer benefits beyond those of other recreational physical activities in different population groups.

## Figures and Tables

**Table 1 ijerph-19-08579-t001:** Participant characteristics.

	*n*	%
Gender		
Boys	3033	44.8
Girls	3708	54.8
Another gender	30	0.4
Ethnicity		
European	4772	70.5
Māori	1052	15.2
Pasifika	210	3.1
Asian	665	9.7
MELAA	86	1.3
Other	16	0.2
Disability status		
Without physical disability	6407	94.6
With physical disability	364	5.4
Social deprivation		
Low deprivation	2827	41.8
Mid deprivation	2714	40.1
High deprivation	1230	18.2

**Table 2 ijerph-19-08579-t002:** Binary logistic regression analyses examining the association between physical activity/organized sport participation and wellbeing.

	Model 1	Model 2	Model 3
	OR (95%CI)		
Active (any physical activity; referent: no physical activity)	3.07 (2.46–3.83)	2.49 (1.97–3.13)	
Meeting physical activity recs (≥420 min/week)	1.85 (1.67–2.04)	1.63 (1.47–1.81)	
Physical activity (hours/week)	1.04 (1.03–1.04)	1.03 (1.02–1.04)	.
Any organized sport (referent: no organized sport)	1.96 (1.77–2.18)	1.78 (1.60–1.98)	1.66 (1.49–1.86)
Organized sport duration (hours/week)	1.12 (1.10–1.13)	1.11 (1.09–1.13)	1.09 (1.07–1.11)

Note. Model 1—no adjustments; Model 2—adjusted for socio-demographic characteristics; Model 3—adjusted for socio-demographic characteristics and total physical activity.

## Data Availability

Publicly available datasets were analysed in this study. This data can be provided on request from research@sportnz.org.nz.

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
