# Peer review of "The Value of Sport: Wellbeing Benefits of Sport Participation during Adolescence"

_ijerph, 2022, doi:10.3390/ijerph19148579_

Round 1

Reviewer 1 Report

The publication is very interesting and valuable, but I think that having such a huge material collected, it would be worth doing more analyzes to show more dependencies, the work would be richer. The same when it comes to the time spent on activity, maybe it would be worth showing how many hours / week is the time that brings special joy. This is a publication has very large group of 6771 young people more analyzes will be interesting.

For activities that they had participated in, participants were asked in what  settings they had participated in (‘in PE or class at school’, ‘in a competition or tournament’, ‘training or practicing with a coach/instructor’, ‘playing or hanging out with family or friends’, ‘playing on my own’, or ‘for extra exercise, training, or practice without a coach or instructor’) –-  I think  that there is a division into very important categories, because e.g. 'playing or hanging out with family or friends could give an individual more happiness than e.g. ‘playing on my own', it could be possible to perform additional analyzes and show which type of activity is still particularly important for young people

In the discussion section, I would suggest writing a few sentences about the influence of physical activity on human physiology, i.e. the release of endorphins. It seems that it cannot omit the discussion of this issue, because movement and physical activity affect the release of endrfins and therefore the player, especially recreational, feels better.

The references section is not prepared according to the guidelines from the journal.

Reviewer 2 Report

  • The authors examined wellbeing benefits of organized sport participation over and above physical activity participation within a specific geographical and sociocultural context.  Justification was provided for studying wellbeing specifically within Aotearoa New Zealand, and for investigating sport and recreational physical activity in terms of potential applications.  Data collection involved  a national sample with publicly available data.  Results are stated clearly.  Recommendations for applications and future research are well-grounded in the findings.  The results pertaining to physical activity and organized sport participation are meaningful, and contribute to a growing body of literature.
  • While the single item well-being measure and the potential lack of cultural representativeness of that item are weaknesses, limitations on methodology and research design are identified and explained overtly.
  • One recommendation for the discussion would be to further connect findings back to the NZ Living Standards Framework, which was introduced in the initial paragraph.

Round 2

Reviewer 1 Report

Accept in present form